# Market Behavior and Evolution of Wealth Distribution: A Simulation Model Based on Artificial Agents

**Andrea Giunta [1], Gaetano Giunta [2,3], Domenico Marino [4] and Francesco Oliveri [1,*]**

[1]  Department of Mathematical and Computer Sciences, Physical Sciences and Earth Sciences, University of Messina, Viale F. Stagno d'Alcontres 31, 98166 Messina, Italy; and.giunta@gmail.com
[2]  Interuniversitary Horcynus Orca Foundation, Capo Peloro, 98164 Messina, Italy; g.giunta@fdcmessina.org
[3]  Community Foundation of Messina o.n.l.u.s., Forte Petrazza, Camaro Sup, 98151 Messina, Italy
[4]  Department of Heritage, Architecture and Urban Planning Mediterranean, University of Reggio Calabria, Salita Melissari, 89124 Reggio Calabria, Italy; dmarino@unirc.it
*   Correspondence: foliveri@unime.it

**Abstract:** The aim of this work is to simulate a market behavior in order to study the evolution of wealth distribution. The numerical simulations are carried out on a simple economical model with a finite number of economic agents, which are able to exchange goods/services and money; the various agents interact each other by means of random exchanges. The model is micro founded, self-consistent, and predictive. Despite the simplicity of the model, the simulations show a complex and non-trivial behavior. First of all, we are able to recognize two solution classes, namely two phases, separated by a threshold region. The analysis of the wealth distribution of the model agents, in the threshold region, shows functional forms resembling empirical quantitative studies of the probability distributions of wealth and income in the United Kingdom and the United States. Furthermore, the decile distribution of the population wealth of the simulated model, in the threshold region, overlaps in a suggestive way with the real data of the Italian population wealth in the last few years. Finally, the results of the simulated model allow us to draw important considerations for designing effective policies for economic and human development.

**Keywords:** artificial agents; market simulation; wealth distribution

## 1. Introduction

The possibility of constructing models suitable to interpret and predict the relationship between wealth distribution and economic dynamics was certainly a central research topic for decades, and has had an important revival in recent years due to numerical simulation studies.

An interesting article by Kuznets in 1955 [1], where the relationship between inequality and economic growth has been extensively investigated, can be considered as the starting point of this kind of analysis. The evaluation question of his studies can be summarized as follows: "Is there a possible relationship between this secular swing in income inequality and the long swing in other important components of growth process?" The answer given to this question by Kuznets, using the empirical data of some countries, is very illuminating, even if it contradicts what traditional growth theories usually highlight. By comparing the level of equality (or inequality), the relationship derived from Solow growth models is substantially denied. Comparing empirical data, it is found, systematically, that the countries that have shown the greatest growth in recent years are those where a higher level of equality has been pursued.

Subsequent checks on empirical data [2–4] confirmed Kuznets' results and questioned the conclusions of the theoretical neoclassical models. In fact, traditional theories relate inequality to growth through an effect on capital accumulation. The reasons for this can be summarized as follows [5]:

- the marginal propensity to save money of the rich people is higher than that of the poor people;
- the indivisibility of the investments and the presence of high sunk costs require a large money availability to start investments.

In conclusion, it would seem that, in this class of models, inequality is a necessary condition for the accumulation of capital, hence, for the economic growth, and consequently redistributive processes.

Although these considerations remain partially valid under certain conditions, they nevertheless give a partial view of the problem, and fail to explain the correlations between equality and growth.

Besides the numerous economic and economic-social studies which provide more complex interpretative keys of the phenomena covered by our work, in recent years, models of statistical mechanics have been developed [1,6–18]; furthermore, many numerical simulations of interacting artificial agents built with the explicit aim of explaining, starting from predictive microscopic models, the evidence obtained through empirical studies, and better understanding the limits of validity of the classic approaches by Solow, have been carried out.

The attempt of this work is, precisely, to simulate the behavior of a market, starting from a simple model of interaction, and thus study the evolution of the distribution of wealth. Our work is in the wake of some foundational studies. First of all, reference should be made to the work of Chakraborti and Chakrabarti [19], or to the most recent studies developed in [20,21], where it has been introduced a simple three-parameter model, called *affine wealth model*, and Ref. [22] where there is an attempt, despite its essentiality, to fit real data with excellent accuracy.

The simulations of this work have the specific objective of studying the economic dynamics related to the evolution of the distribution of wealth as a function of the initial allocations, in order to understand how the latter affect their evolution. The study could provide interesting indications for designing both efficient and fair socio-economic policies.

The urgency of building predictive models suitable to give indications on how welfare systems can be able to redistribute initial stocks of goods is evident to us if we take into account that, in the last few decades, world inequalities have dramatically grown; for instance, the recent explosive growth of China has been accompanied by a strong growth of private capital, concentrated in attractors of wealth, and a worrying decline of the public capital (see, for instance, [6,23,24]). The output data of the simulations developed here will be finally compared with the real distribution data of financial wealth in Italy and may allow us to estimate how the Covid-19 pandemic could strongly influence the "type" of the socio-economic dynamics if effective and prompt actions devoted to the redistribution of wealth are not adopted.

The plan of the paper is as follows. In Section 2, we introduce the basic ingredients of our model and specify the rules governing its evolution; in Section 3, we present and discuss the results of some numerical simulations obtained varying the parameters herein involved; in particular, we identify the critical parameter responsible for the different evolutions we can obtain. Finally, Section 4 contains some concluding remarks.

## 2. The Model

In this section, we briefly describe the ingredients of our model of market and the *rules* governing its evolution. We assume to have a market where $N$ agents $A_\alpha$ ($\alpha = 1, \ldots, N$) exchange financial activities between them, having as a constraint only their actual currency endowment. The $N$ agents can be either producers or consumers. Each agent $A_\alpha$ has an initial supply $m_\alpha^0$ of money; we impose as a constraint that this initial value remains below a parameter denoted $b_{max}$. We also assume that in the market $M$ different goods/services do exist and can be exchanged; the unitary price $p_i$ ($i = 1, \ldots, M$) of each good/service is set randomly in a predefined range $(p_{min}, p_{max})$ during the initialization phase; this cost will no longer be changed during the iterations. Only one type of good/service that will not

change during the iterations is randomly associated with each economic agent; in addition, an initial quantity $q_i^0$ of the asset in a predefined range $(q_{min}, q_{max})$ is randomly attributed to each economic agent.

Each agent can use her money stock to buy goods/services from other agents, obtain money by selling the actual stock of owned goods/services, and produce new goods/services to sell on the market once she is going to run out of the initial allocation stock and she wants to restore the stock of goods/services owned. The sale of the produced goods/services is the only way to acquire additional money (besides the initial allocation) necessary for further exchanges.

Once the market and its actors have been initialized, we let the system to evolve through a sequence of transactions according to the following rules:

1.   two agents, $A_\alpha$ and $A_\beta$, are randomly selected: they represent the seller and the buyer, respectively (each node of the network can assume the role of seller or buyer);
2.   the quantity of good/service in a predetermined range $(m_{min}, m_{max})$, fixed in the initialization phase, is randomly chosen;
3.   if the total cost of the good/service involved in the transaction is in the buyer's availability and if the quantity of goods extracted is in the seller's availability, the transaction is accepted, otherwise, it is rejected; goods/services associated with a successful transaction are considered consumed and therefore *removed* from the system;
4.   the increase in money associated with the seller's monetary deposit and the related decrease in goods/services in the seller's availability stock, as well as the decrease in the buyer's monetary deposit, are recorded;
5.   to avoid that the system inevitably stops, the model foresees that economic agents periodically produce a well-defined quantity of goods/services: more in detail, after a transaction, if the actual quantity of the goods/services in the seller's deposit turns out to be less than 1/4 of the initial one, then the seller replenishes her stock of goods/services to her initial quantity by means of her labor.

As fas as the rules of the model are concerned, some comments are in order. In a more realistic model, each buyer (labeled $\alpha$) associated with a good/service (labeled $i$) in order to process successfully a transaction has to satisfy the so called budget constraint,

$$W_\alpha = m_\alpha + f(L_\alpha)p_i + q_i p_i$$

where the production function $f(L_\alpha)$ is determined only by the labor factor $L_\alpha$, $m_\alpha$ is the current amount of money, and $q_i$ the current stock of goods/services with unitary cost $p_i$. Nevertheless, our model is such that each transaction has to satisfy the much simpler currency constraint. Moreover, the production function is simulated by automatically replenishing the amount of goods/services of an agent when her stock of goods/services becomes less than 1/4 of the initial one. A less simplified model is currently being investigated and will be presented in a forthcoming paper.

All the transactions are stored in a database in order to have the detailed evolution of the market. In the simulations, we fix the parameters involved in the model as follows. Various numerical simulations have been carried out choosing increasing numbers of economic agents ($N = 50, 100, 200, 400, 800$), whereas the number of distinct good/services is taken as $M = N/2$.

These values may seem small; indeed, the number of possible interactions at each iteration, from which the algorithm selects one randomly, equal to $N(N-1)$, is sufficiently high. By considering that the simulations are developed up to $20N^2$ successful transactions, also to verify the non-ergodic conditions of the dynamics, the model is expected to well simulate a macroscopic system that can profitably be compared with a real market.

The choices of the parameters associated with the results we will show in the sequel are as follows. The unit cost of the asset, associated with each node during the initialization phase, will be in the range $(1, 10)$, the quantity of goods/services associated with each node during the initialization phase is randomly selected in the range $(100, 200)$; the quantity

of good/service involved in a generic transaction will be randomly chosen in the range $(10, 20)$; the initial amount of money of the economic agents is randomly chosen in the range $(b_{max}/2, b_{max})$. These choices guarantee that, at the beginning of the evolution, there is no dominant position; in fact, the maximum supply of goods/services and money of the economic agents is not greater than the double of the minimum of corresponding supply. These parameters are similar to those used in a preliminary analysis by one of the authors (A.G), except for the number of nodes $N$, which was 30 in that preliminary study. Note that all the values for quantities exchanged, prices of the goods/services, initial money and good/service stocks are natural numbers chosen in their predefined ranges.

It is worth observing that our numerical experiments clearly showed that these initialization parameters do not affect the quality and the features of the phase diagram that the system manifests as the value of parameter $b_{max}$ changes; therefore, only the parameter $b_{max}$, which is the maximum amount of money that can be attributed during the initialization phase to each node of the system, seems to play a relevant role. The simulations of the present study were, therefore, developed as the value of the parameter $b_{max}$ is varied (from 200 to 10,000). Finally, moving from the value of $N = 30$, used in the preliminary study, to higher values of $N$, allowed us to verify possible scaling effects on the system.

The model is micro-founded, self-consistent, predictive, and normative. The study is developed by analyzing the dynamics of the distribution of wealth among the $N$ agents, after checking the saturation of the scale effects. Therefore, the numerical simulation solves a simple model of economy with a finite number of artificial economic agents (the nodes of the network) based on a random exchange system. Ultimately, it is a system of heterogeneous artificial agents, made distinguishable by the initialization procedure, able to exchange goods/services and money exclusively within the simulated economic network. We can imagine the simulated system as a set of economic agents who, interacting with each other, acquire or lose money and in a complementary and opposite way an amount of goods/services.

The model is a closed system in relation to the number of economic agents involved and the total wealth of the system; nevertheless, it turns out to be an open model concerning the quantities of goods/services exchanged during the simulations of the system, that is, during the numerical iterations. The analysis was focused on the study of the evolution of the distribution of wealth among the artificial agents along the simulations as the initial conditions change.

## 3. Numerical Results

The objective of the analysis was to study how much the predictive solutions of the simulated model can provide useful interpretative keys of some basic mechanisms of a real economic system, and therefore how they can provide general guidelines to construct policy hypotheses. All numerical simulations have been performed by using Matlab™ (2020a release).

The distribution of wealth among the economic agents changes during the evolution of the system; therefore, we compute the coefficient of variation (i.e., the square root of the variance divided by the mean) of the amounts of money of the different agents as transactions go on; it is commonly accepted that the coefficient of variation fulfills the requirements for a measure of economic inequality [13].

The plots in Figure 1 show the evolution of the coefficient of variation with increasing iterations of the different simulations carried out respectively for $b_{max} = 200, 500, 1000, 2000, 4000, 6000, 8000, 10,000$, when the number of economic agents is 50.

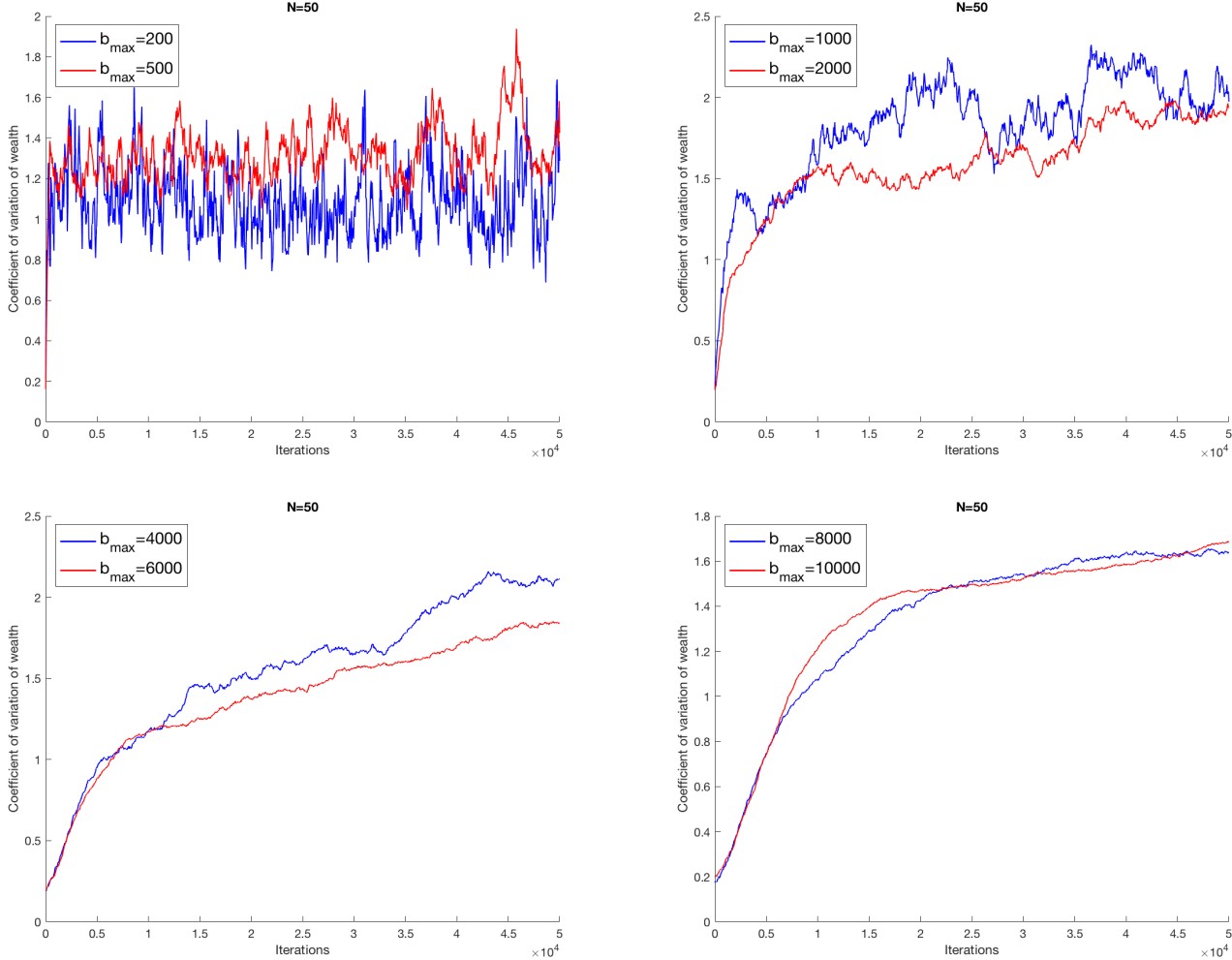

**Figure 1.** Plot of the coefficient of variation of the amount of money of the economic agents as transactions go on for $N = 50$ and $b_{max} = 200, 500$ (**top left**), $b_{max} = 1000, 2000$ (**top right**), $b_{max} = 4000, 6000$ (**bottom left**), $b_{max} = 8000, 10,000$ (**bottom right**).

Despite the essentiality of the model, the simulations show a complex and non-trivial behavior. First of all, two classes of solution are highlighted, two phases, separated by a threshold value $b_{max}$ in a neighborhood of the region $500 < b_{max} < 1000$. For $b_{max}$ below this interval, the economic system evolves according to a typical random walk. For $b_{max} > 1000$, the coefficient of variation assumes an almost growing monotonous evolution, thus showing a progressive concentration of wealth in dominant attracting nodes. In this phase of solutions, the economic dynamics will gradually be trapped. The situation does not change if we increase the number of economic agents; we report in Figure 2 the results for $N = 100$, and in Figure 3 those for $N = 200$; moreover, the same qualitative behavior is exhibited for higher values of $N$. It is worth being stressed that the features of the output of the model are preserved for a higher number of successful transactions, as shown in Figure 4, where we take a system with 400 agents among which we consider up to $16 \times 10^6$ successful transactions.

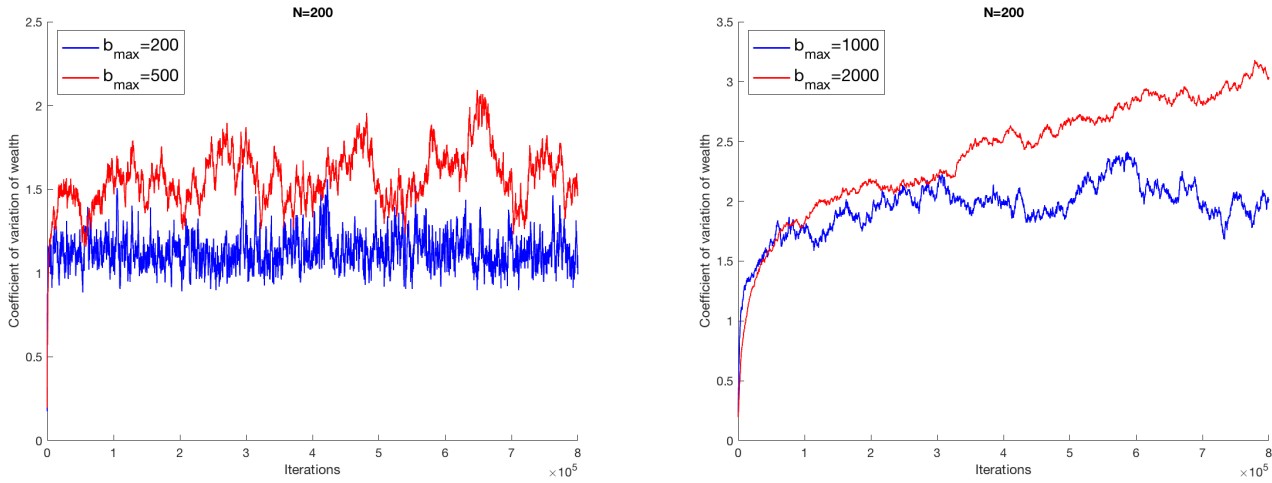

**Figure 2.** Plot of the coefficient of variation of the amount of money of the economic agents as transactions go on for $N = 100$ and $b_{max} = 200, 500$ (**top left**), $b_{max} = 1000, 2000$ (**top right**), $b_{max} = 4000, 6000$ (**bottom left**), $b_{max} = 8000, 10,000$ (**bottom right**).

**Figure 3.** *Cont.*

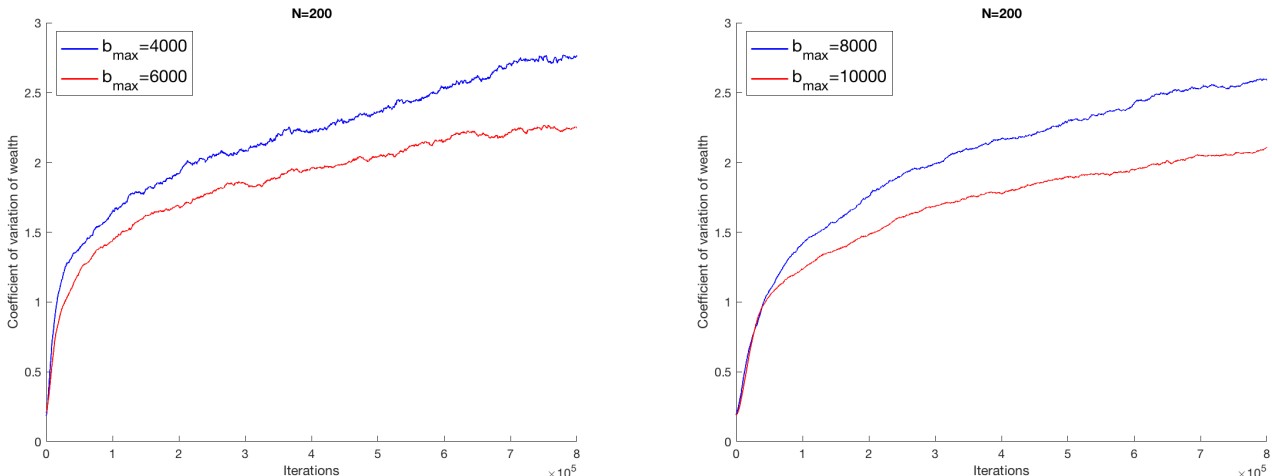

**Figure 3.** Plot of the coefficient of variation of the amount of money of the economic agents as transactions go on for $N = 200$ and $b_{max} = 200, 500$ (**top left**), $b_{max} = 1000, 2000$ (**top right**), $b_{max} = 4000, 6000$ (**bottom left**), $b_{max} = 8000, 10{,}000$ (**bottom right**).

**Figure 4.** Plot of the coefficient of variation of the amount of money of the economic agents as transactions go on for $N = 400$ and $b_{max} = 200, 500$ (**top left**), $b_{max} = 1000, 2000$ (**top right**), $b_{max} = 4000, 6000$ (**bottom left**), $b_{max} = 8000, 10{,}000$ (**bottom right**). The number of successful transactions is $100N^2$.

To prove that the critical parameter affecting the numerical output of the model is $b_{max}$, in Figure 5, we report the trends of the coefficient of variation of the amount of money of the economic agents as transactions go on for $N = 400$, in the case where initially all economic agents have the same amount of money, and of monetary value of the goods/services (both equal to $b_{max}$). In such a way, all agents enter the market with the same conditions, and inequality arises only as a consequence of the random fluctuations induced by transactions.

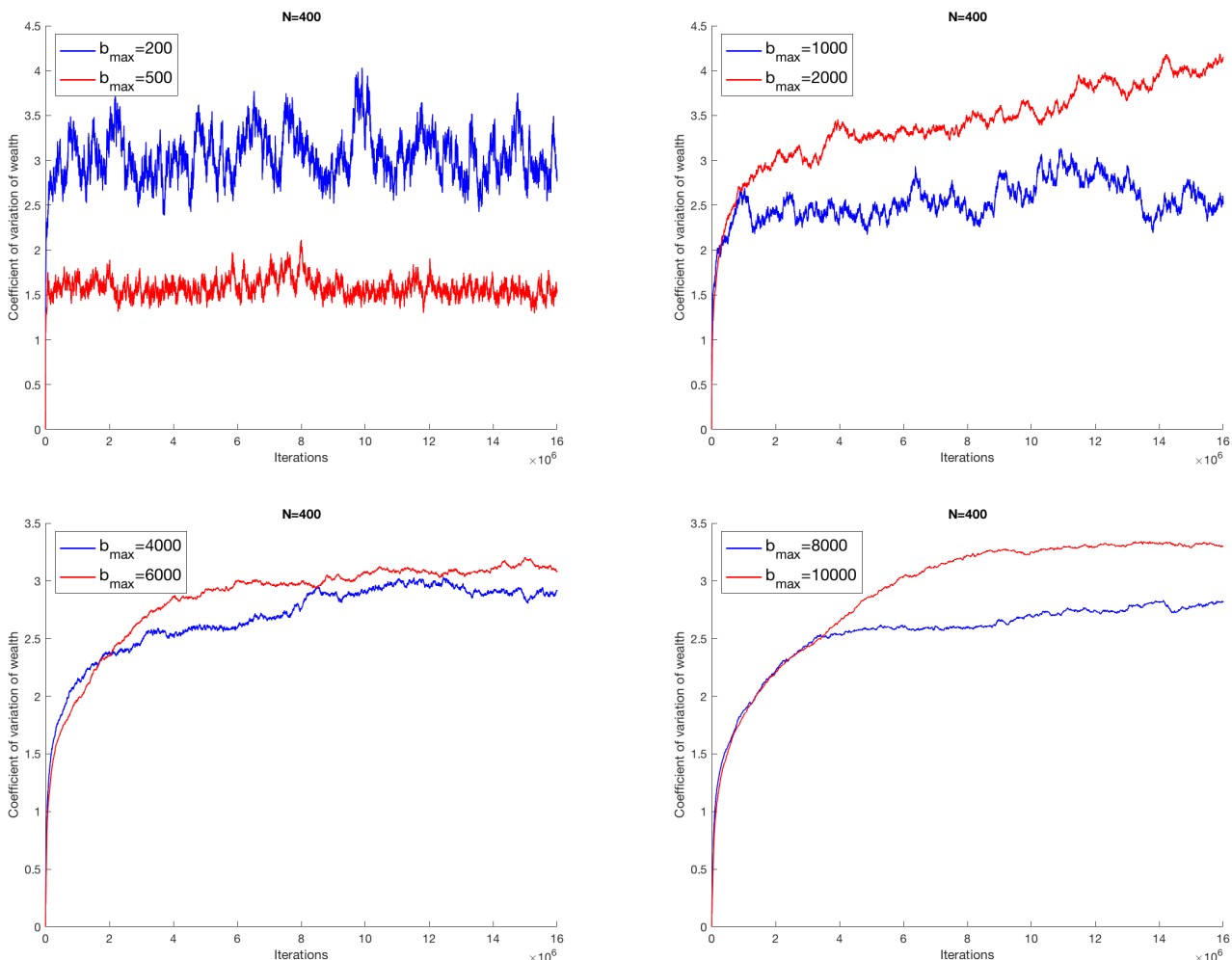

**Figure 5.** Plot of the coefficient of variation of the amount of money of the economic agents as transactions go on for $N = 400$ and $b_{max} = 200, 500$ (**top left**), $b_{max} = 1000, 2000$ (**top right**), $b_{max} = 4000, 6000$ (**bottom left**), $b_{max} = 8000, 10,000$ (**bottom right**). The number of successful transactions is $100N^2$. All agents enter the market with the same amount of money and the same money equivalent of goods/services.

The existence of two distinct macroscopic phases is consistent with the aforementioned independently developed studies [20–22]. These latter simulations, although carried out in a microscopic model much different from the one proposed here, and therefore analyzed with the variation of other parameters, are based on simple random interactions and show two distinct evolutionary stages of the economic systems. The analysis in the supercritical phase ($b_{max} > 1000$) confirms the non-ergodicity of the model. When the initial conditions, determined randomly, can take on highly unequal features, the system retains memory during future iterations and traps itself progressively without being able to explore the entire space of possible solutions. Figure 6 shows the Gini index at the end of the sequence of transactions for different values of $N$ as a function of $b_{max}$. The trend confirms the conclusions previously described.

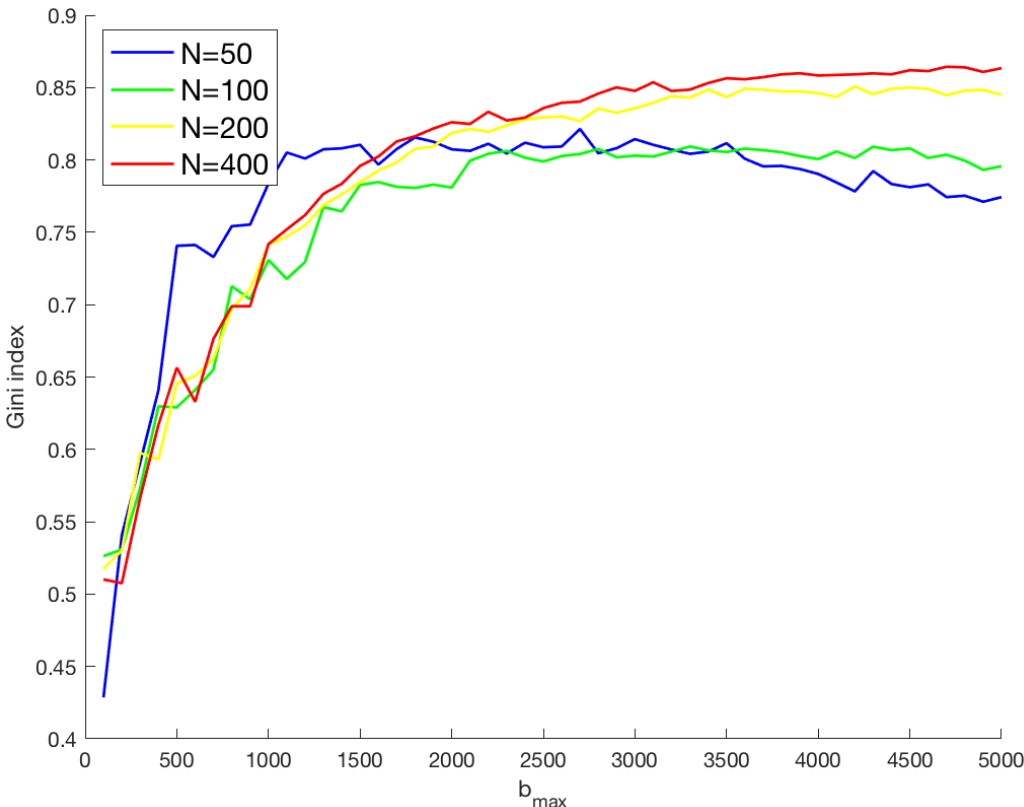

**Figure 6.** Plot of the Gini index at the end of the sequence of the $20N^2$ transactions vs. the value of $b_{max}$ for different values of the number $N$ of the economic agents.

A more detailed analysis of the distribution of wealth shows results of great interest when compared with empirical quantitative studies developed in [25] on probability distributions of wealth and incomes in the United Kingdom and the United States. The plot in Figure 7 shows the cumulative percentage of the population (i.e., economic agents) as a function of the level of wealth, as they emerge from the different simulations carried out when $N = 800$ with $b_{max} = 500, 1000$, which are, respectively, the lower and upper bounds of the threshold region.

For $b_{max}$ not exceeding the value 500, the cumulative percentage of wealth assumes the Boltzmann-Gibbs exponential form. When $b_{max}$ becomes greater than 1000, in the plots, two functional forms of distribution emerge: the Boltzmann–Gibbs exponential form for low levels of wealth, the Pareto polynomial one for higher values of wealth. This behavior is progressively structured and stiffens in simulations for higher values of $b_{max}$, where the system is progressively trapped by the emergence of dominant nodes. The functional forms assumed by the distribution of wealth between the nodes of the model for values of $b_{max}$ close to the upper bound of the threshold region are consistent with the real data fit analyzed in [25]. The model suggests that, in the long run (huge number of iterations in the simulations), the system is statistically less rich but more egalitarian with respect to the initial distribution of resources, provided that $b_{max}$ is below the critical threshold. If, on the contrary, the initial conditions can generate significant wealth polarity (above a certain threshold), during the iterations that characterize the simulations, growing dominant attractors emerge, which end up creating traps that limit the dynamics of the economic system. The foregoing results suggest to us to compare the real data of a country's distribution of wealth with the results of the simulations at the threshold value. The comparison between the decile distribution of wealth population of simulations (in the case $N = 800$, $b_{max} = 1000$) and of the Italian population in 2016 (source: Bank of Italy) is reported in Figure 8. The results appear extremely encouraging.

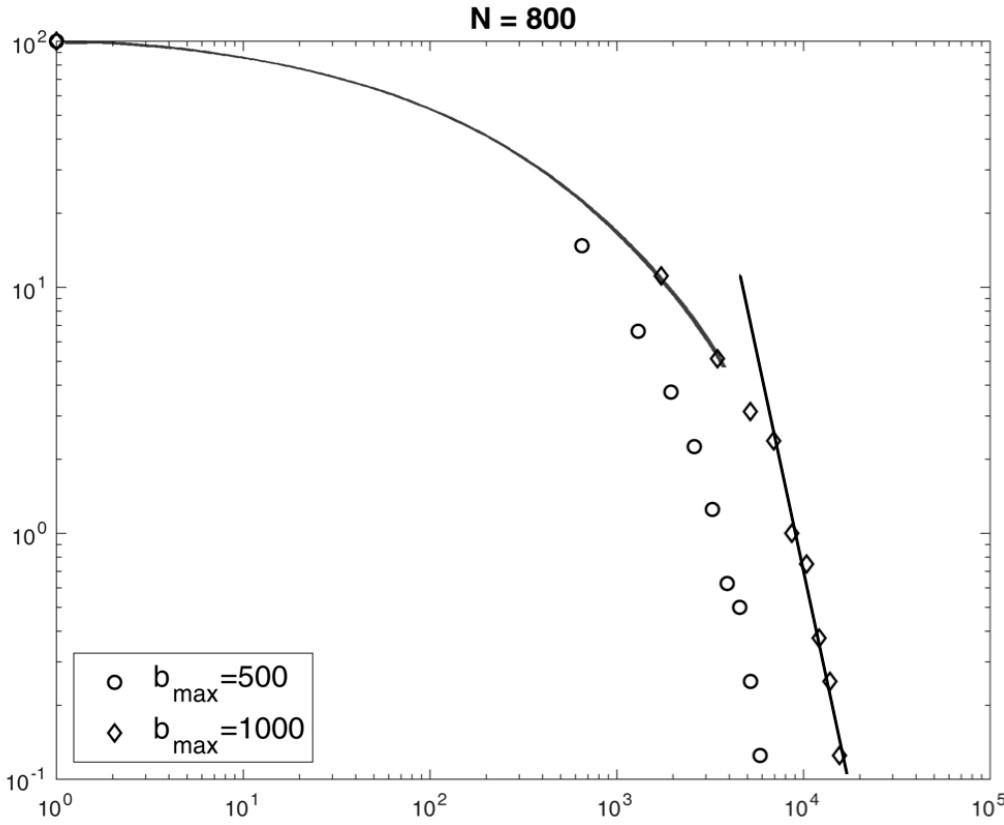

**Figure 7.** Log-log plot of cumulative percentage of population (800 nodes) according to the level of wealth after the sequence of transactions in the cases $b_{max} = 500, 1000$.

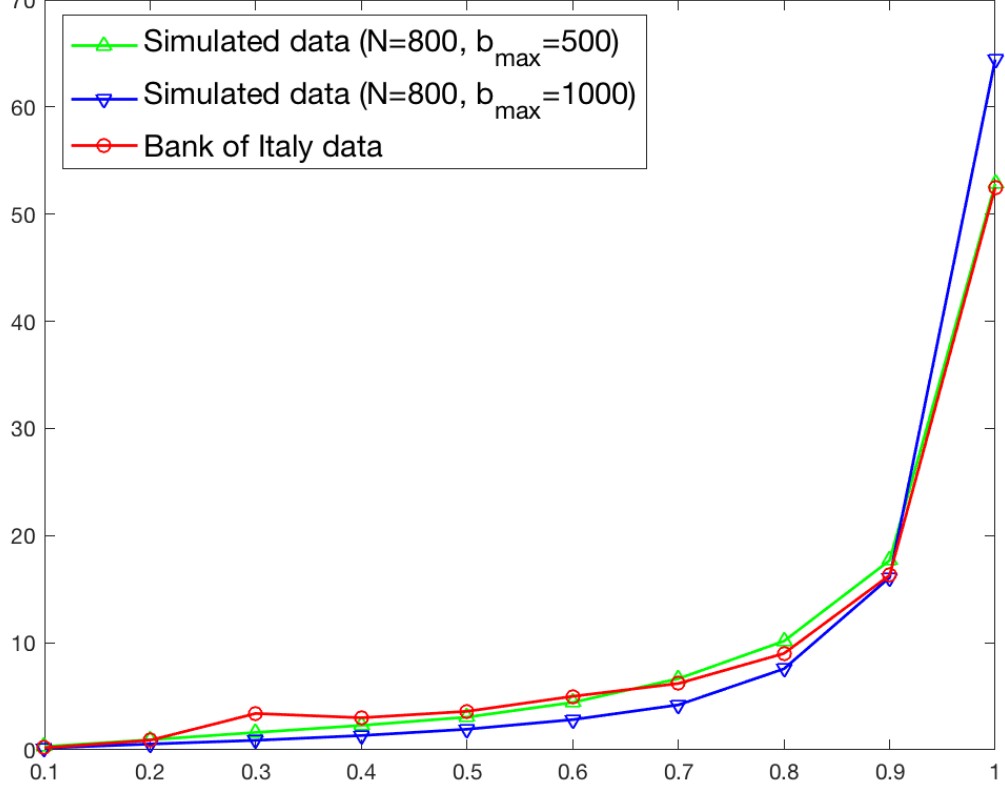

**Figure 8.** Comparison between the decile distribution of wealth in the simulation with $b_{max} = 500, 1000$, and the decile distribution of real Italian financial wealth in 2016 (source: Bank of Italy).

Finally, the value of the Gini index, output of the simulations for the value of $b_{max} = 500, 1000$, lower and upper bound of the threshold region, are also in a suggestive agreement with that emerging from the real distribution of Italian financial wealth, which has entered the threshold region in the last few years (see Table 1).

**Table 1.** Gini index at the end of the sequence of simulated transactions for $N = 800$ and $b_{max} = 500, 1000$ (on the left), and (on the right) Gini index of *per capita* net wealth distribution in Italy in the period 1991–2016 (source: Bank of Italy).

| Simulations ($N = 800$) | | Year | Gini Index |
|---|---|---|---|
| | Gini Index | 1991 | 0.593 |
| | | 1993 | 0.616 |
| | | 1995 | 0.611 |
| | | 1998 | 0.635 |
| | | 2000 | 0.641 |
| $b_{max} = 500$ | 0.6260 | 2002 | 0.621 |
| $b_{max} = 1000$ | 0.7477 | 2004 | 0.617 |
| | | 2006 | 0.631 |
| | | 2008 | 0.632 |
| | | 2010 | 0.644 |
| | | 2012 | 0.657 |
| | | 2014 | 0.635 |
| | | 2016 | 0.635 |

If we accept the results of the simulation that place the real distribution of wealth in Italy at the critical region of the phase diagram, it follows that an exogenous shock able to determine a *jump*, in relatively short times, in the radicalization of the distribution of wealth could lead to the launch of supercritical economic dynamics. The latest report of the Bank of Italy estimates a worrying jump in inequalities due to the socio-economic crisis caused by the Covid-19 pandemic. The Bank of Italy, in its Annual Report, in fact, estimates that, in the first quarter of 2020, 20% of households with lower incomes have suffered a double reduction in its income compared to the loss suffered by families belonging to the highest fifth. The inequality in the distribution of labor income, measured by the Gini index, in the first quarter of 2020 has already been increased by about two percentage points, to 37%, say, the maximum value since 2009, the first year of the great economic crisis. For now, the impact is calmed by the social safety nets, but, *in the medium term, there is a risk that the Covid-19 emergency will accentuate the inequalities*. The well-founded concern that, following the Covid-19 crisis, a new condition of restart will occur, characterized by a distribution of wealth of a supercritical type, requires the urgent implementation of redistributive stock policies.

## 4. Discussion and Conclusions

Final or better interpretative considerations of the results presented in this paper cannot be separated from a fundamental premise: the simulation model starts from very little restrictive hypotheses. In fact,

- the agents operate in conditions of total symmetry of information and capabilities;
- the interaction is free and is independent of the proximity;
- no agent has a market power, the iterations are random, and there is no hypothesis of economic rationality.

We could speak of conditions that approach the ideal model of perfect competition; however, this would be outside of any hypothesis of economic utilitarianism. On the other hand, however, in the view of the results obtained, the model, despite the simplicity of the hypotheses on which it is based, seems to contain the founding nuclei underlying the

dynamics of wealth allocation. The simulations, in fact, deny, under certain conditions, the neoclassical Solow-like approaches and exhibit very complex behaviors. When the algorithm of initialization of the dynamics of exchanges determines too unequal initial conditions of wealth, the supercritical phase, the simulated economic processes progressively become trapped. The model returns a critical proximity threshold necessary for the very survival of economic dynamics, which, therefore, unlike neoclassical thought, does not perform, in such cases, any redistributive function. If the system, even in the presence of hypotheses that approach the perfect competition, manages to develop a macroscopic phase, for values of $b_{max}$ higher than the threshold region, in which, as the iterations proceed, an ever greater number of agents remains excluded from the transactions, and the system becomes more and more unequal, it is expected that such behaviors will be strengthened if conditions closer to reality are introduced, in the absence of strategically targeted corrective welfare policies—for example, introducing hypotheses of market power, asymmetric information, interactions between first-neighbors, or taking into account that economic inequalities are strongly correlated and mutually amplified by social inequalities [7], recognition inequalities [11], and the level of people's capabilities [12,26].

The simulations developed in this work may allow for studying how the initial stocks are correlated with the dynamics of the flows, and clearly highlighting how policy aimed at preventing, and/or re-activating and freeing the socio-economic trajectories of territories and communities at risk of falling and/or which have fallen below the poverty trap line, must be able to redistribute wealth. This study clearly tells us that only if the flows are based on stocks that have a sufficient level of proximity can we develop dynamics that are both efficient and fair. The awareness explained above, in which inequalities are inseparably multidimensional, suggests that we re-interpret the threshold as an access capability, a sort of stock of capabilities. From what has been said so far, it is clear how welfare policies aimed at redistributing and/or creating stocks of material goods, relational goods, knowledge, rather than flows—as hypothesized in [22]—have to be considered necessary investments, not costs, to guarantee not only justice but even economic development. Precisely for this reason, in the socio-economic dynamics of a country, they need to be placed as external constraints on any logic of economic utilitarianism.

**Author Contributions:** Conceptualization, methodology, and software: A.G. and F.O.; validation: G.G., and D.M; writing—original draft preparation: G.G. and D.M.; writing—review and editing: F.O. All authors have read and agreed to the published version of the manuscript.

**Funding:** This research received no external funding.

**Acknowledgments:** This work is supported by Interuniversitary Horcynus Orca Foundation, Community Foundation of Messina o.n.l.u.s., Department of Heritage, Architecture and Urban Planning Mediterranean of the University of Reggio Calabria, and Department of Mathematical and Computer Sciences, Physical Sciences, and Earth Sciences of the University of Messina.

**Conflicts of Interest:** The authors declare no conflict of interest.

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
