# Peer review of "Market Behavior and Evolution of Wealth Distribution: A Simulation Model Based on Artificial Agents"

_mca, doi:10.3390/mca26010012_

Round 1

Reviewer 1 Report

The paper basically states, that parameter bmax affects the speed and extent of inequality growth in the surveyed economy, which represents the maximum possible endowment of monetary units to each agent at initialisation phase. I wish the mechanism of stated influence was described more thoroughly, probably, illustrated graphically with some scheme. The overall quality of the paper meets necessary standards and can be published if the uttered comments are properly addressed.

Author Response

We thank the referee for his/her comments and criticisms that enabled us to better clarify some aspects and improve the quality of the paper.

We provided the results (figure 5 in the revised version) of a new simulation of 16 millions transaction with 400 economic agents where all agents enter the market with the same amount of money and the same money equivalent of goods/services (both equal to max). The observed evolution qualitatively does not change, so proving the relevant role played by this parameter (lines 172-177 of the revised version of the paper).

Reviewer 2 Report

The authors introduce a micro-founded model of transactions with heterogeneous agents with quenched properties.  Monte Carlo studies are carried out, and results indicating distinct macroscopic phases and loss of ergodicity are presented.  Work of this sort is of great interest currently, and the model and results in this paper are well presented.  I recommend publication in its present form

Author Response

We thank the referee for his/her appreciation of the paper. 

Nevertheless some revisions have been made in order to clarify the basic ingredients of the model and the rules governing the evolution of the system: lines 93-94, 96-97, 98-108 and 169-177 of the revised version.

We added two more figures (4 and 5) to support our conclusions.

Reviewer 3 Report

The description of the model in Section 2 appears to be missing essential details. Steps 1-4 make it clear how transactions between agents occur in the model. However, you do not describe how stocks of good are replenished. What do agents do with the goods/services that they buy?

You state that “it is worth of being observed that our numerical experiments clearly showed that these initialization parameters do not affect the quality and the features of the phase diagram that the system manifests … only the parameter b_{max} seems to play a relevant role.” If this is the case, then it suggests to me that the model is more complicated than necessary to produce the desired result. Though incomplete, the model description appears to suggest just two essential processes:

(1) The transfer of wealth from agent A to agent B in an economic transaction.

(2) The growth of wealth by whatever process new goods/services are produced (which is not described in the model).

Could the model be reduced to just these two processes, without the explicit exchange of goods/services for liquid money and the later (I presume) exchange of money for new stocks of goods and services? Some explanation of should be given concerning the role of distinguishing goods/services from liquid wealth given that the distinction does not appear to affect the model outcome.

I am not wholly convinced that changing b_{max} fundamentally changes the dynamics of the model. Rather, I expect that the rate at which inequality forms depends on the size of the initial disparity in wealth across the population. By increasing b_max, you increase the initial disparity due to larger range (b_max/2,b_max), which results in a more rapid formation of the pareto-exponential distribution. If the simulations with lower values of b_max were run for a larger number of iterations, I expect that you would see a dynamic similar to what is seen over a shorter time scale with a larger value of b_max. This should be easy to test with longer simulation runs of the small b_max cases.

If this is true, it would fundamentally (I believe) alter the conclusion that there is a threshold value of b_max, and so it is (in my view) very important to convincingly demonstrate that the long run dynamics are what the authors’ expect through simulation experiments with many more iterations or by an appropriate mathematical analysis. I had sought to do this experiment myself, but found that the model description was not adequate to reproduce the reported simulation results.

Author Response

We thank the referee for his/her comments and criticisms that enabled us to better clarify some aspects and improve the quality of the paper.

In the revised version, we described the process of producing new goods/services.The growth of wealth by whatever process new goods/services are produced (lines 93-94, 96-97 and 98-108).

We distinguished goods/services from liquid wealth just to simulate what occur in real situations when goods/services are obtained by exchanging money. Moreover, we measured the wealth of the agents by means of their money stock., and use for transactions only a currency constraint (lines 104-105). 

As far as the role of b_{max} in changing the dynamics of the model, we added a new simulation (the plots in figure 5 of the revised version) where all economic agents enter the market with the same amount of money and the same monetary value of stocks of goods/services; nevertheless, inequality still arises (as a consequence of the random fluctuations induced by transactions), and the qualitative trend already described is preserved (lines 172-177).

In order to show that for a higher number of transactions the observed trends are unchanged, we included a new figure (fig. 4 of the revised version) by considering as an example 400 agents and up to 16 millions of successful transactions (lines 169-171).

Round 2

Reviewer 3 Report

This revision addressed my concerns except for one issue in the model description. Are the ranges for quantities exchanged, price, initial monies, etc. real valued or natural numbers? This information should be added to complete the description of the model.

Author Response

In lines 128-130 of the revised manuscript, we specified that the quantities exchanged, the prices of the goods/services, and the initial money and good/service stocks, are natural numbers in their predefined ranges.